# Exploring the use of self-management strategies in survivors of critical illness: A scoping review protocol

Anastasia N. L. Newman[1]*, Christopher Farley[1], Shannon McKenney[2], Jenna Smith-Turchyn[1]

**1** School of Rehabilitation Science, Faculty of Health Science, McMaster University, Institute for Applied Health Sciences (IAHS) Building, Hamilton, Canada, **2** Sepsis Canada, Hamilton, Canada

☯ These authors contributed equally to this work.
* newmanan@mcmaster.ca

## Abstract

### Introduction

Survivors of critical illness often experience lasting physical, cognitive, and mental health impairments that persist beyond hospital discharge. While self-management strategies have demonstrated benefits in many chronic conditions, their application in survivors of critical illness remains limited. With no current synthesis of the literature, there is an increasing need to understand and optimize self-management strategies in this population.

### Materials and methods

This is a protocol for a scoping review which will systematically map the literature on self-management strategies used with adult survivors of critical illness. Following Joanna Briggs Institute guidelines and reported according to PRISMA-ScR, this review will include peer-reviewed studies involving ICU survivors (≥24-hour admission) that employ self-management strategies, such as education, peer support, and coping skills, and report patient-centered outcomes. A comprehensive search of MEDLINE, Embase, Emcare, CINAHL, PsycINFO, and AMED will be conducted from inception. Two independent reviewers will screen studies, extract data, and resolve disagreements with a third reviewer, if needed. Data will be summarized with descriptive statistics and narrative synthesis.

### Conclusion

The results of this scoping review will summarize the available evidence on the use of self-management strategies in survivors of critical illness, identify knowledge gaps, and inform future research and clinical practice. A manuscript will be submitted for publication in a peer-reviewed journal and presented at critical care conferences.

**Data availability statement:** No datasets were generated or analysed during the current study. All relevant data from this study will be made available upon study completion.

**Funding:** The author(s) received no specific funding for this work.

**Competing interests:** The authors have declared that no competing interests exist.

## Introduction

Intensive care units (ICU) provide life-saving treatment to critically ill patients. Prior to the development of the ICU from the poliomyelitis epidemic of the 1950s in Denmark, the clinical course of an acutely ill patient involved physiological decline and eventual death. Since the inaugural ICU, advances in medical interventions and health care technologies have led to more effective management of patients with critical illness and improved survival [1]. However, ICU survivors often experience Post-Intensive Care Syndrome (PICS), a condition characterized by new or worsened impairments in physical function, cognitive and/or mental health that can last well beyond hospital discharge [2]. These deficits, which include muscle weakness, reduced exercise capacity, frailty, fatigue, cognitive impairment, anxiety and depression, often stem from the rapid muscle wasting and dysfunction acquired during their ICU stay and can persist for months or years [2–4]. Landmark longitudinal research identified only partial recovery of physical function capacity among survivors of acute respiratory distress syndrome, with mean six-minute walk test scores remaining well below age- and sex-matched norms at one- and five-year follow-up [5,6]. The persistent nature and prevalence of these symptoms often lead to the development PICS [7]. The challenges associated with PICS are often compounded by limited access to post-ICU outpatient clinics, which provide structured, multidisciplinary support, but are few in number and often inaccessible to many patients [8–10]. In this context, self-management strategies offer a promising means for survivors of critical illness to manage symptoms, optimize quality of life and support functional recovery after hospital discharge.

Self-management refers to the active participation of individuals to manage their symptoms, treatments and implement lifestyle changes associated with a chronic health condition [11–13]. Self-management interventions are described as interventions that provide patients with the skills necessary to manage their condition, aimed at improving self-health behaviour, emotional and social wellbeing, developing self-management skills, and increasing the individual's responsibility for healthcare decisions [14]. In collaboration with healthcare providers, many of these interventions foster patient problem-solving and decision-making skills, which improve quality of life and reduce resource utilization, including rehospitalization [11,13,14]. Self-management strategies have been successfully implemented in a variety of chronic health conditions, including patients with diabetes, chronic obstructive pulmonary disease, arthritis, or cardiovascular disease [13]. Evidence suggests that, compared to usual care, self-management strategies promote higher quality of life, self-efficacy, and symptom control in people living with chronic illnesses [13,15]. Despite widespread use among people with other chronic conditions, the implementation of self-management strategies for survivors of critical illness and those with PICS remains limited [16]. Emerging evidence suggests that tailored self-management interventions, including education, peer support, and behavioural strategies, can help ICU survivors manage persistent challenges such as fatigue and impaired functional status, and may offer a promising framework for recovery post ICU [17].

To date, there is no comprehensive synthesis of the literature on the role of self-management strategies in survivors of critical illness. With the growing body of

evidence, and increased survival of ICU patients, demand will likely increase for effective self-management strategies to address the long-term physical, cognitive, and mental health impairments experienced by survivors. Therefore, a scoping review is warranted to systematically map the existing literature, identify knowledge gaps, and inform future research and clinical practice on the optimization of self-management interventions for survivors of critical illness.

### Research question

1. What self-management strategies are being utilized with survivors of critical illness/PICS?

## Materials and methods

This review will follow the standard scoping review methodology described by the Joanna Briggs Institute (JBI) and will be reported according to the Preferred Reporting Items for Systematic Reviews and Meta-Analyses extension for Scoping Reviews (PRISMA-ScR) [18,19]. Our review has been registered a priori on Open Science Framework (OSF) (OSF | Exploring the use of self-management strategies in survivors of critical care: A scoping review).

### Eligibility criteria

The development of our inclusion criteria was guided by the Population, Concept, Context (PCC) framework as per the JBI guidelines (Table 1) [19]. Peer reviewed full-text papers will be eligible for inclusion if they: [1] include adult survivors of critical illness (i.e., admitted to ICU for ≥ 24 hours) with or without a diagnosis of PICS, [2] incorporate one or more self-management strategies, including but not limited to education, peer-support, medication management, goal setting, and coping strategies, and [3] discuss the impact of the self-management strategies on recovery outcomes, such as physical functioning, quality of life, self-efficacy, mental health wellbeing, cognitive functioning, fatigue, return to work, and healthcare utilization. Reasons for article exclusion include: [1] published in a language other than English, French, and Portuguese and [2] grey literature, conference abstracts, letters to the editor, systematic and narrative reviews.

### Search strategy

Search strategies will be developed in consultation with a health sciences librarian using the search terms "intensive care", "self-management", and "post-hospital" and their synonyms or associated terms using Boolean operators. A preliminary search strategy is available in Appendix 1. The search strategy will be adapted for each database and validated by testing whether it identifies three relevant articles [17,20,21] meeting our inclusion criteria previously identified through Google scholar within each database.

### Information sources

A comprehensive search of the following databases will be performed from their inception to identify peer-reviewed studies: Ovid MEDLINE, Embase, Emcare, AMED, PsycINFO, and EBSCOhost CINAHL. References of included studies will be searched for other potentially relevant studies.

**Table 1. Population, Concept, Context Statement.**

| Population | Community-dwelling adult survivors of critical illness (i.e., admitted to ICU for ≥ 24 hours) with or without a diagnosis of PICS |
|---|---|
| Concept | Use of self-management strategies |
| Context | Community setting. Any English, French, or Portuguese peer-reviewed published literature. No limitations on location, sex, or race. No limitation on publication timeline. |

## Data screening and extraction

Two reviewers will independently screen titles, abstracts, and full-text articles, with any disagreements resolved through discussion. If consensus cannot be achieved, a third reviewer will adjudicate. Prior to commencing title/abstract screening and full text selection, calibration exercises involving 10 articles will be conducted to ensure consistency between reviewers. Screening will be facilitated using the Covidence web-based platform (Covidence, Boston, Massachusetts, USA), while data extraction will be completed using predefined tables in Microsoft Excel. A pair of reviewers will independently extract data and reconcile discrepancies to reach agreement. The following data will be extracted from each included full text: [1] study identification (i.e., title, first author, year of publication, journal title, country of origin, trial registration number, study design, purpose/objectives, and funding sources), [2] sample characteristics (i.e., sample size, age, sex, ICU length of stay, diagnoses upon ICU admission, comorbidities), [3] description of self-management strategies employed, and [4] outcomes assessed (e.g., quality of life, physical function, health care utilization). We expect screening to begin in February 2026 data extraction occurring in March 2026.

## Data analysis

Descriptive statistics and narrative analyses will be used to summarize and present data. Descriptive statistics, including counts, percentages, means (standard deviations), and medians (1st and 3rd quartiles), will be utilized to analyze numerical data. A PRISMA flow diagram will be used to illustrate the review process. Extracted data will be presented throughout the manuscript and in both tabular and graphical formats, as appropriate. Narrative analysis will be used to synthesize the self-management strategies identified in the included studies, with results organized according to the PCC framework and summarized to highlight patterns in strategy type, delivery mode, and context of application among survivors of critical illness.

## Dissemination of results

The results of this scoping review will be submitted for presentation at relevant critical care conferences. A manuscript will be written and submitted for consideration of publication in a peer-reviewed journal. We expect to submit our completed manuscript for publication in May 2026.

## Significance

Admission to critical care is not benign and survivors often experience significant impairments that extend beyond hospital discharge. The use of self-management strategies is well supported in a multitude of chronic health conditions and are beginning to be implemented for patients with PICS. This scoping review will synthesize the available evidence on the use of self-management strategies with survivors of critical illness. Results may help guide the development of post-ICU self-management programs aimed at improving outcomes for people with PICS.

## Supporting information

**S1 Appendix. Preliminary Search Strategy.**
(DOCX)

**S2 Appendix. PRISMA-P Checklist.**
(DOCX)

## Acknowledgments

The authors would like to thank Ms. Neera Bhatnagar, Health Sciences Librarian at McMaster University, for her assistance with designing the search strategies.

## Author contributions

**Conceptualization:** Anastasia N.L. Newman, Shannon McKenney, Jenna Smith-Turchyn.

**Data curation:** Anastasia N.L. Newman, Christopher Farley, Shannon McKenney, Jenna Smith-Turchyn.

**Formal analysis:** Anastasia N.L. Newman, Christopher Farley, Shannon McKenney, Jenna Smith-Turchyn.

**Investigation:** Anastasia N.L. Newman.

**Methodology:** Anastasia N.L. Newman, Christopher Farley.

**Project administration:** Anastasia N.L. Newman, Jenna Smith-Turchyn.

**Resources:** Anastasia N.L. Newman.

**Software:** Anastasia N.L. Newman.

**Supervision:** Anastasia N.L. Newman, Jenna Smith-Turchyn.

**Writing – original draft:** Anastasia N.L. Newman.

**Writing – review & editing:** Anastasia N.L. Newman, Christopher Farley, Shannon McKenney, Jenna Smith-Turchyn.

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
