## [Decision Letter · Decision Letter 0]

25 Mar 2026

Dear Dr. Newman,

Thank you for submitting your manuscript to PLOS ONE. After careful consideration, we feel that it has merit but does not fully meet PLOS ONE’s publication criteria as it currently stands. Therefore, we invite you to submit a revised version of the manuscript that addresses the points raised during the review process.

As the corresponding author, your ORCID iD is verified in the submission system and will appear in the published article. PLOS supports the use of ORCID, and we encourage all coauthors to register for an ORCID iD and use it as well. Please encourage your coauthors to verify their ORCID iD within the submission system before final acceptance, as unverified ORCID iDs will not appear in the published article. *Only* the individual author can complete the verification step; PLOS staff the individual author can complete the verification step; PLOS staff the individual author can complete the verification step; PLOS staff the individual author can complete the verification step; PLOS staff *cannot* verify ORCID iDs on behalf of authors.verify ORCID iDs on behalf of authors.verify ORCID iDs on behalf of authors.verify ORCID iDs on behalf of authors.

We look forward to receiving your revised manuscript.

Kind regards,

Katherine Demi Kokkinias, Ph.D.

Staff Editor

PLOS One

Journal Requirements:

4. Please include captions for your Supporting Information files at the end of your manuscript, and update any in-text citations to match accordingly. Please see our Supporting Information guidelines for more information: http://journals.plos.org/plosone/s/supporting-information....

Additional Editor Comments:

Please note that we have only been able to secure a single reviewer to assess your manuscript. We are issuing a decision on your manuscript at this point to prevent further delays in the evaluation of your manuscript. Please be aware that the editor who handles your revised manuscript might find it necessary to invite additional reviewers to assess this work once the revised manuscript is submitted. However, we will aim to proceed on the basis of this single review if possible.

The reviewer raised several that need attention. They request additional information on methodological aspects of the study (such as the inclusion criteria, the search strategy and description of the narrative analysis). There were also suggestions to improve the description of the self-management interventions and correct typographical errors.

Could you please revise the manuscript to carefully address the concerns raised?

Reviewer's Responses to Questions

**Comments to the Author**

1. Does the manuscript provide a valid rationale for the proposed study, with clearly identified and justified research questions?

Reviewer #1: Yes

2. Is the protocol technically sound and planned in a manner that will lead to a meaningful outcome and allow testing the stated hypotheses?

Reviewer #1: Yes

3. Is the methodology feasible and described in sufficient detail to allow the work to be replicable?

Reviewer #1: Yes

4. Have the authors described where all data underlying the findings will be made available when the study is complete?

The PLOS Data policy requires authors to make all data underlying the findings described in their manuscript fully available without restriction, with rare exception, at the time of publication. The data should be provided as part of the manuscript or its supporting information, or deposited to a public repository. For example, in addition to summary statistics, the data points behind means, medians and variance measures should be available. If there are restrictions on publicly sharing data—e.g. participant privacy or use of data from a third party—those must be specified.requires authors to make all data underlying the findings described in their manuscript fully available without restriction, with rare exception, at the time of publication. The data should be provided as part of the manuscript or its supporting information, or deposited to a public repository. For example, in addition to summary statistics, the data points behind means, medians and variance measures should be available. If there are restrictions on publicly sharing data—e.g. participant privacy or use of data from a third party—those must be specified.

Reviewer #1: Yes

5. Is the manuscript presented in an intelligible fashion and written in standard English?

Reviewer #1: Yes

You may also provide optional suggestions and comments to authors that they might find helpful in planning their study.

Reviewer #1: Dear authors,

Thank you for the opportunity to read and review your scoping review protocol. An important topic for a group of clients that deserve more attention. The introduction clearly describes the importance of the review, as well as the need for more insight into self-management strategies. The method is eligible, clearly described according to the steps of the JPI. We realize that you already started with your study. However, we have a few small questions and considerations regarding the protocol:

o Typo in abstract, line 31: menta health should be mental?

In the introduction, you mention critical illness and PICS, but PICS does not appear in the inclusion criteria. Is that still an option to consider? If yes, this might also lead to a refinement in the aim/question.

In lines 82-84, you describe self-management interventions. This still seems to be more focused on illness and medical management in the description, but self-management is a concept which is broader. Consider also describing and including attention for role and emotional management.

Analysis, line 165-166: Consider to conduct a narrative analysis also about the self-management strategies, as the aim is to have a deeper understanding about the strategies. When you only analyze the PCC, it might give limited understanding. Furthermore, how would you carry out the narrative analysis? In the screening and extraction phase, the steps are described more clearly

o The language indicated in the PCC framework is only English; more languages are mentioned in the text. What choice do you make in this regard?

o In the search strategy only critical illness is mentioned, why do you not include critical condition or critical disease as a search term?

For the data extraction, funding sources are included, why do you find this important?

Good luck with your scoping review.

.

Reviewer #1: **Yes:** Ton SatinkTon SatinkTon SatinkTon Satink

---

## [Author Response · Author response to Decision Letter 1]

30 Mar 2026

March 30, 2026

Dr. Katherine Demi Kokkinias, PhD

Staff Editor

PLOS One

Dear Dr. Kokkinias,

RE: Exploring the use of self-management strategies in survivors of critical illness: A scoping review protocol

Project ID: PONE-D-25-63311

Principal Investigator: Dr. Anastasia Newman, MSc(PT), MSc(RS), PhD

Thank you for reviewing our protocol manuscript, entitled “Exploring the use of self-management strategies in survivors of critical illness: A scoping review protocol”. We are grateful for opportunity to revise our protocol so that it meets the standards for publication at PLOS One. We thank the Reviewer and Editor for their time and feedback. You will find our responses to the reviewer’s comments below.

Reviewer 1 Comments:

1. Typo in abstract, line 31: menta health should be mental?

RESPONSE: Thank you for identifying this spelling error. It has been corrected.

2. In the introduction, you mention critical illness and PICS, but PICS does not appear in the inclusion criteria. Is that still an option to consider? If yes, this might also lead to a refinement in the aim/question.

RESPONSE: Thank you for identifying this omission. While not all survivors of critical illness will meet the criteria for PICS, we have added PICS to the PCC table and the research question.

3. In lines 82-84, you describe self-management interventions. This still seems to be more focused on illness and medical management in the description, but self-management is a concept which is broader. Consider also describing and including attention for role and emotional management.

RESPONSE: Thank you for this comment and for noting this lack of inclusive definition. We have added “emotional and social wellbeing”, as stated in the Effing et al reference.

4. Analysis, line 165-166: Consider to conduct a narrative analysis also about the self-management strategies, as the aim is to have a deeper understanding about the strategies. When you only analyze the PCC, it might give limited understanding. Furthermore, how would you carry out the narrative analysis? In the screening and extraction phase, the steps are described more clearly.

RESPONSE: Thank you for this feedback. We have expanded this section to provide more detail about the narrative analysis: “Narrative analysis will be used to synthesize the self-management strategies identified in the included studies, with results organized according to the PCC framework and summarized to highlight patterns in strategy type, delivery mode, and context of application among survivors of critical illness.”

5. The language indicated in the PCC framework is only English; more languages are mentioned in the text. What choice do you make in this regard?

RESPONSE: Thank you for identifying this error. We are including papers published in English, French, and Portuguese due to the language fluencies of our study team. The omission of “French” and “Portuguese” in the PCC Table was incorrect. We have added the following to the PCC table: “Any English, French, or Portuguese peer-reviewed published literature.”

6. In the search strategy only critical illness is mentioned, why do you not include critical condition or critical disease as a search term?

RESPONSE: Thank you for this suggestion. In all our search strategies, we included the subject heading “critical illness” and exploded the term to ensure comprehensiveness. The terms “critical condition” and “critical disease” are not subject headings and can only be searched as keywords. Our search strategy was reviewed and edited by a health sciences librarian and was validated to ensure it could identify 3 key articles.

7. For the data extraction, funding sources are included, why do you find this important?

RESPONSE: Thank you for this comment. Funding source was included as a data extraction item because it contributes to transparency and allows readers to consider potential influences on study conduct and reporting. In line with scoping review objectives, this item is being captured descriptively rather than analytically.

All documents submitted and uploaded have an updated version number and version date as requested. We appreciate the opportunity to further strengthen our application and hope these changes will allow for publication in PLOS One.

Thank you for reviewing our resubmission.

Sincerely,

Dr. Anastasia Newman, MSc(PT), MSc(RS), PhD

Assistant Professor, School of Rehabilitation Science

McMaster University

newmanan@mcmaster.ca

905-745-0666

---

## [Decision Letter · Decision Letter 1]

6 Apr 2026

Exploring the use of self-management strategies in survivors of critical illness: A scoping review protocol

PONE-D-25-63311R1

Dear Dr. Newman,

We’re pleased to inform you that your manuscript has been judged scientifically suitable for publication and will be formally accepted for publication once it meets all outstanding technical requirements.

Kind regards,

JONATHAN BAYUO, PhD

Academic Editor

PLOS One

Additional Editor Comments (optional):

Reviewers' comments:

Reviewer's Responses to Questions

**Comments to the Author**

1. Does the manuscript provide a valid rationale for the proposed study, with clearly identified and justified research questions?

Reviewer #1: Yes

2. Is the protocol technically sound and planned in a manner that will lead to a meaningful outcome and allow testing the stated hypotheses?

Reviewer #1: Yes

3. Is the methodology feasible and described in sufficient detail to allow the work to be replicable?

Reviewer #1: Yes

4. Have the authors described where all data underlying the findings will be made available when the study is complete?

The PLOS Data policy requires authors to make all data underlying the findings described in their manuscript fully available without restriction, with rare exception, at the time of publication. The data should be provided as part of the manuscript or its supporting information, or deposited to a public repository. For example, in addition to summary statistics, the data points behind means, medians and variance measures should be available. If there are restrictions on publicly sharing data—e.g. participant privacy or use of data from a third party—those must be specified.requires authors to make all data underlying the findings described in their manuscript fully available without restriction, with rare exception, at the time of publication. The data should be provided as part of the manuscript or its supporting information, or deposited to a public repository. For example, in addition to summary statistics, the data points behind means, medians and variance measures should be available. If there are restrictions on publicly sharing data—e.g. participant privacy or use of data from a third party—those must be specified.

Reviewer #1: Yes

5. Is the manuscript presented in an intelligible fashion and written in standard English?

Reviewer #1: Yes

You may also provide optional suggestions and comments to authors that they might find helpful in planning their study.

Reviewer #1: Thank you for your revision and using the feedback for refinement of your protocol. I am looking forward to the review!

.

Reviewer #1: No

---

## [Editor Report · Acceptance letter]

PONE-D-25-63311R1

PLOS One

Dear Dr. Newman,

I'm pleased to inform you that your manuscript has been deemed suitable for publication in PLOS One. Congratulations! Your manuscript is now being handed over to our production team.

Kind regards,

on behalf of

Dr. JONATHAN BAYUO

Academic Editor

PLOS One